# Cytoplasmic Kinase Network Mediates Defense Response to *Spodoptera litura* in Arabidopsis

**DOI:** 10.3390/plants12091747

**Published:** 2023-04-24

**Authors:** Yoshitake Desaki, Minami Morishima, Yuka Sano, Takuya Uemura, Ayaka Ito, Keiichirou Nemoto, Akira Nozawa, Tatsuya Sawasaki, Gen-ichiro Arimura

**Affiliations:** 1Department of Biological Science and Technology, Faculty of Advanced Engineering, Tokyo University of Science, Tokyo 125-8585, Japan; 2Iwate Biotechnology Research Center, Kitakami 024-0003, Japan; 3Proteo-Science Center, Ehime University, Matsuyama 790-8577, Japan

**Keywords:** *Arabidopsis thaliana* (Arabidopsis), PBL27, CRK2, HAK1, *Spodoptera litura*, ethylene

## Abstract

Plants defend against folivores by responding to folivore-derived elicitors following activation of signaling cascade networks. In Arabidopsis, HAK1, a receptor-like kinase, responds to polysaccharide elicitors (Frα) that are present in oral secretions of *Spodoptera litura* larvae to upregulate defense genes (e.g., *PDF1.2*) mediated through downstream cytoplasmic kinase PBL27. Here, we explored whether other protein kinases, including CPKs and CRKs, function with PBL27 in the intracellular signaling network for anti-herbivore responses. We showed that CRK2 and CRK3 were found to interact with PBL27, but CPKs did not. Although transcripts of *PDF1.2* were upregulated in leaves of wild-type Arabidopsis plants in response to mechanical damage with Frα, this failed in CRK2- and PBL27-deficient mutant plants, indicating that the CRK2/PBL27 system is predominantly responsible for the Frα-responsive transcription of *PDF1.2* in *S*. *litura*-damaged plants. In addition to CRK2-phosphorylated ERF13, as shown previously, ethylene signaling in connection to CRK2-phosphorylated PBL27 was predicted to be responsible for transcriptional regulation of a gene for ethylene response factor 13 (ERF13). Taken together, these findings show that CRK2 regulates not only ERF13 phosphorylation but also PBL27-dependent de novo synthesis of ERF13, thus determining active defense traits against *S*. *litura* larvae via transcriptional regulation of *PDF1.2*.

## 1. Introduction

In response to herbivory, plants activate signaling cascade networks for emergent defense responses that bring about herbivore resistance. Initially, to recognize the herbivore damage, plants perceive elicitors, such as herbivore-associated molecular patterns (HAMPs), secreted by feeding herbivores concomitantly with physical damage, to trigger vigorous defense responses against herbivore pests [1,2].

In the model plant *Arabidopsis thaliana* (Arabidopsis), herbivore damage signal-associated receptor-like kinase (HAK1), localized along the plasma membrane, responds specifically to the polysaccharide elicitor fractions (FrA or Frα [3]) that are present in oral secretions (OS) of *Spodoptera litura* larvae, leading to activation of intracellular signaling for defense response [4]. This intracellular signaling cascade is mediated through the downstream cytoplasmic kinase PBS1-like 27 (PBL27) [4], as shown likewise for chitin elicitor receptor kinase 1 (CERK1)-LysM receptor kinase 5 (LYK5), a receptor complex that recognizes the microbe-associated chitin elicitor to transmit phosphorylation signals to PBL27 and receptor-like cytoplasmic kinase VII-4 (RLCK VII-4) members for plant immune responses [5,6,7,8]. The HAK1/PBL27 system elicits de novo production of ethylene and the resultant robust upregulation of transcripts of defense genes (e.g., *plant defensin 1.2* [*PDF1.2*]), thus determining active defense traits against *S*. *litura* larvae [4].

In addition to the HAK1/PBL27 system, other cytoplasmic kinases should act concomitantly or independently in Arabidopsis during plant damage by *S*. *litura* larvae [2]. For instance, calcium-dependent protein kinases (CPK3 and CPK13) have been shown to mediate the phosphorylation of the heat shock factor HsfB2a (Hsf22) in order to upregulate the transcription of *PDF1.2* [9]. CPKs constitute a large family of serine/threonine protein kinases in plants and play multi-functional roles in the activation of mitogen-activated protein kinases (MAPKs) [10], nicotinamide adenine dinucleotide phosphate oxidases (NADPH oxidases) [11,12], and transcription factors [9,13] to orchestrate intracellular signaling networks upon herbivore and pathogen attack [14].

Moreover, a member of the CPK-related protein kinases (CRK2) has been shown to phosphorylate tyrosine (Tyr) residues of a subset of transcription factors, including herbivory-responsive ethylene response factor 13 (ERF13) and RAP2.6 (ERF108) in Arabidopsis, which bind to genomic GCC boxes and activate defense genes including *PDF1.2* [15,16]. Likewise, CRK3 plays key roles in Tyr-phosphorylation of WRKY14, which binds to genomic W boxes and activates defense genes [15]. Such Tyr-phosphorylations are responsible for the nuclear localization, DNA-binding, and transactivation of transcription factors, as shown for CjWRKY1, which is involved in the biosynthesis of benzylisoquinoline alkaloids in *Coptis japonica* [17]. Notably, CRK2 is multifunctional, phosphorylating not only transcription factors but also GARU, gibberellin receptor RING E3 ubiquitin ligase, which promotes ubiquitin-dependent proteasome degradation of gibberellin receptor (GID1) in Arabidopsis [18].

Based on these findings, there is no doubt that the intracellular kinases PBL27, CPKs, and CRKs are responsible for defense responses in Arabidopsis, but whether and how they engage in cross-talk are unknown. Thus, to further understand the functioning of the intracellular signaling network, we explored the polysaccharide elicitor-responsive transcript of *PDF1.2* in order to examine its possible regulation through cross-talk between PBL27 and other cytoplasmic protein kinases in response to *S*. *litura* attack in Arabidopsis.

## 2. Results

### 2.1. Molecular Interaction of PBL27 with Cytoplasmic Kinases

First, to screen CRKs and CPKs that interact with PBL27, we assessed in vitro interaction between the biotinylated recombinant PBL27 protein and FLAG-conjugated cytoplasmic kinase proteins (CRK2, CRK3, CPK3, and CPK13) using the AlphaScreen system. The resultant luminescence intensities showed that PBL27 protein interacted strongly with CRK2 and CRK3 and weakly with CPK3 and CPK13 when compared with *Escherichia coli* dihydrofolate reductase serving as the control protein (Figure 1A). Moreover, co-immunoprecipitation assays confirmed in vivo interactions between HA-tagged PBL27 and FLAG-tagged CRK2 and CRK3 in *Nicotiana benthamiana* leaf cells following their transient expression (Figure 1B). In contrast, such in vivo interactions were not observed between HA-tagged PBL27 and FLAG-tagged CPK3 or CPK13 (Appendix A). Therefore, to further study the PBL27-interacting cytoplasmic kinases found here, hereafter, we focused on CRKs.

### 2.2. Central Role of the PBL27/CRK2 System in S. litura Elicitor Response

Next, to further screen cytoplasmic kinases that play a significant role in *S*. *litura* polysaccharide elicitor (Frα [3,4])-responsive transcription of the representative defense gene *PDF1.2*, leaves of Arabidopsis wild-type (WT) and its T-DNA insertion mutant lines of PBL27 (*pbl27*), CRK2 (*crk2*), and CRK3 (*crk3*) were subjected to mechanical damage (MD) with application of Frα. *PDF1.2* transcript levels in WT leaves were increased in response to MD + Frα but not MD alone (Figure 2), as shown previously [4]. These responses were not observed in *pbl27* or *crk2* leaves, but were observed in *crk3* leaves, indicating that the *S*. *litura*-induced *PDF1.2* expression is dependent on PBL27/CRK2.

### 2.3. Selective Phosphorylation Ability Comparison between PBL27 and CRK2

Next, to assess whether PBL27 and CRK2 phosphorylate each other, we performed phosphorylation assays using phosphate affinity electrophoresis (phos-tag sodium dodecyl sulfate (SDS)-polyacrylamide gel electrophoresis (PAGE)). Concomitant expression with WT CRK2 (CRK2^WT^) in *N*. *benthamiana* leaf cells resulted in phosphorylation of the kinase domain-mutant (kinase dead [KD]) of PBL27 (PBL27^KD^), whose lack of kinase activity has been shown previously [6], according to the phosphorylation assays (Figure 3). In contrast, when WT PBL27 was concomitantly expressed with KD of CRK2 (CRK2^KD^ [15,16]) in *N*. *benthamiana*, CRK2^KD^ was scarcely phosphorylated (Figure 3).

### 2.4. Promotion of PDF1.2 Transcription with the PBL27/CRK2/ERF13 System

Regarding the fact that PBL27 is a substrate of CRK2 (see Figure 3), it has been shown that CRK2 also phosphorylates a transcriptional factor, ERF13, leading to activation of the defense gene *PDF1.2* in Arabidopsis leaves [15]. Moreover, given that the 5′ flanking region of the *PDF1.2* gene (1 kb) (PDF1.2P) contains a potential ERF-binding motif (GCC box [AGCCGCC]) at 311 bp (base pairs) upstream of the start codon, it was predicted that ERF13 serves as potent activator of PDF1.2P. In fact, when *ERF13* was coexpressed with a *firefly luciferase (Fluc)* reporter gene under the control of PDF1.2P (PDF1.2P: *Fluc*) in protoplasts prepared from Arabidopsis WT leaves, the levels of Fluc activity produced due to the transformed reporter construct was dramatically increased compared to the level in protoplasts expressing PDF1.2P: *Fluc* alone (Figure 4A). The Fluc activity levels were not as dramatically increased in protoplasts prepared from *pbl27* or *crk2* leaves, indicating that PBL27 and CRK2 are responsible for activation of ERF13 (Figure 4A). However, PDF1.2P-promoted Fluc activity levels with *ERF13* expression were increased when *CRK2* was coexpressed, but not when *PBL27* was coexpressed, compared to those with *ERF13* alone, in WT leaf protoplasts (Figure 4B). We therefore hypothesized that CRK2 is directly committed to ERF13 activation but PBL27 is not.

### 2.5. Ethylene Signaling for Transcriptional Activation of ERF13

Based on the findings regarding a possible indirect involvement of PBL27 in the transcriptional regulation of *PDF1.2* (Figure 4B), we next focused on transcriptional regulation of *ERF13*. First of all, we assessed whether the *ERF13* transcript level was responsive to Frα in a PBL27-dependent manner. For this, leaves of Arabidopsis WT and *pbl27* mutant plants were subjected to MD with application of Frα. Increased *ERF13* transcript level was elicited in response to MD + Frα in WT leaves but less in *pbl27* mutant leaves (Figure 5A), indicating that *ERF13* transcription was responsive to Frα in a PBL27-dependent manner. Moreover, given the fact that PBL27 has been shown to be involved in de novo ethylene biosynthesis in Arabidopsis in response to Frα [4], we next assessed whether ethylene signaling was involved in the regulation of *ERF13* transcription. For this purpose, leaves of WT Arabidopsis plants were treated with ethephon, a chemical replacement for ethylene treatment [19]. The transcript level of *ERF13* was upregulated by treatment with ethephon, compared to that in untreated leaves (Figure 5B), indicating that *ERF13* transcription was responsive to ethylene.

## 3. Discussion

In Arabidopsis, CRK2 is multi-functional, controlling an array of regulatory molecules, including transcriptional factors involved in JA signaling [15,16], as well as GARU [18], in various signaling cascades. In addition to these previous findings, here we provided new insight into the complex function of CRK2, which coordinates with PBL27 for resultant transcriptional activation of *PDF1.2* in response to a polysaccharide elicitor (Frα) during *S*. *litura* attack (summarized in Figure 6). PBL27 is a member of receptor-like cytoplasmic kinases [20], which act as a major class of signal transmitting proteins to serve for cellular responses to elicitors such as HAMPs [4] and microbe-associated molecular patterns (MAMPs) [5,6]. Therefore, the CRK2/PBL27 complex may commonly function in HAMP and MAMP responses.

Notably, following phosphorylation of PBL27, ethylene signaling may serve a major intermediate signaling with de novo synthesis of ERF (Figure 5). In connection to this fact, it has been shown that PBL27 regulates the MAPK kinase kinase 5 (MAPKKK5), in the MAPK cascade [5], and that pathogen-responsive MAPKs (MPK3 and MPK6) regulate ethylene production [21]. Moreover, the 1-amino-cyclopropane-1-carboxylic acid synthase (ACS) is one of the rate-limiting enzymes for ethylene production, and ACS isomers (ACS2 and ACS6) have been shown to serve as substrates of MPK3 and MPK6 [22,23]. In the sequence of reactions in such signaling pathways, phosphorylation of ACS2/ACS6 by MAPKs is certainly required to stabilize the ACS protein and its activation, leading to de novo ethylene synthesis in Arabidopsis during biotic and abiotic stress responses [22,23]. Collectively, these facts show that it is indeed conceivable that the PBL27-mediated activation of ethylene-signaling controls ethylene-responsive transcription of *ERF13*. In other words, CRK2 is dual-functional by modulating ERF13 protein function directly and *ERF13* transcriptional activity indirectly, eventually leading to transcriptional activation of *PDF1.2* for anti-herbivore defense (Figure 6). However, the molecular mechanism by which the phosphorylation of ERF13 by the complex of CRK2 and PBL27 activates ERF13 transcription remains unclear. To reveal how protein phosphorylation exerts significant effects on the specific transcription machinery, further analysis of the interaction between the complex and the ERF13 promoter using electrophoretic mobility shift assays, etc., will be required.

It should be noted that transient expression of *ERF13* in protoplasts prepared from the *pbl27* mutant resulted in reduced activation levels of PDF1.2P compared to that in WT protoplasts (Figure 4A). Moreover, given that PBL27 was not directly involved in ERF13 transactivation (Figure 4B), PBL27 might contribute to not only MAPK/ethylene-dependent regulation of the *ERF13* transcript level but also to transactivation of ERF13 indirectly. PBL27 has been shown to phosphorylate not only MAPKKK5, as described [5], but also slow-type (S-type) anion channels [24]. Alternatively, for the transactivation of *ERF13* by PBL27, additional enhancer molecule(s) may be required.

Unlike CRK2, CRK3 is not likely to interact strongly with PBL27 (Figure 1) and to be aggressively involved in the Frα-mediated defense response (Figure 2). This possibility is in accord with the finding that ABA-responsive WRKY14, a CRK3 substrate [15], does not control *PDF1.2* transcription (Appendix A). However, given the fact that CRK3 is responsible for transcriptional regulation of *PDF1.2* during *S*. *litura* attack [15], herbivore danger signals other than Frα, e.g., oral bacteria in *S*. *litura* larvae, may be more responsible for mediating the effects of CRK3 during herbivory. This is in accord with the fact that ABA signaling is enhanced by oral bacteria such as *Staphylococcus epidermidis* [25]. Alternatively, since WRKYs also play a central role in the environmental stress response towards, e.g., draught and heat stresses [26], CRK3 may be involved in responses to not only herbivory but also environmental stresses.

Unlike CRKs, CPK3 and CPK13 might not be predominantly involved in Frα response during *S*. *litura* attack, as they did not interact with PBL27 (Appendix A). Alternatively, since calcium influx mediated via CNGC19, a calcium-permeable channel [27], would be able to activate those CPKs [28], this calcium-dependent signaling may work for signal transduction in a manner independent of the Frα response via the CRK/PBL27 system. Namely, the HAMP-mediated signal network may be sophisticatedly constructed by both the independent actions and cross-talks of multiple protein kinases in plants in response to herbivory.

## 4. Materials and Methods

### 4.1. Plants and Elicitor

*Arabidopsis thaliana* (Arabidopsis) ecotypes Col-0 and its T-DNA insertion mutant lines (*pbl27* [GK_958D06], *crk2* [SALK_090938C], *crk3* [SALK_128719C]) were grown in soil in climate-controlled rooms at 22 ± 1 °C with a photoperiod of 12 h (80 µE m^–2^ s^–1^). Likewise, the potted *Nicotiana benthamiana* plants were grown at 24 ± 1 °C with a photoperiod of 16 h (80 µE m^–2^ s^–1^). The individual plants were grown in single plastic pots. The potted Arabidopsis and *N*. *benthamiana* plants were grown for 4 weeks and 6 weeks, respectively, and were used for subsequent analyses.

Eggs of *Spodoptera litura* (Fabricius) were obtained from Sumika Technoservice Co. Ltd. (Takarazuka, Japan). They were incubated in a climate-controlled room at 24 ± 1 °C with a photoperiod of 16 h. The hatched larvae were reared on artificial diet (Insecta LFS, Nihon Nosan Kogyo Ltd., Tokyo, Japan) in a plastic container (0.9 L) with a mesh-covered lid. Feces in the plastic case were removed and a piece of artificial diet was added 3 times a week. OS was collected from third or fourth instar larvae of *S*. *litura* (about a week after hatching) using a glass capillary tube (Hirschmann Laborgeräte GmbH and Co. KG, Eberstadt, Germany). Briefly, the collected OS was stored at −20 °C until use. OS was passed through a column (1.5 cm × 80 cm) packed with Bio-Gel P-2 resin (Bio-Rad, Hercules, CA, USA) to collect FrA. FrA was subsequently passed through a column (2.0 cm × 45 cm) packed with Bio-Gel P-10 resin (Bio-Rad) to collect Frα. Frα was lyophilized and then dissolved with 750 µL of 10 mM MES buffer (pH 6.0) for assays (2-fold concentrated), according to the method described previously [4].

Frα was diluted 5 fold with 10 mM MES buffer (pH 6.0). MD was performed with stainless steel needles on 3 leaves of an individual Arabidopsis plant. Approximately 30 MD spots were made per leaf. The Frα was immediately applied on an MD spot (approximately 1 µL per spot). Treatment of MD leaves with MES buffer served as a control. Arabidopsis plants were placed in an air-tight container and sprayed with 1mL of 10 mM ethephon solution (Tokyo Chemical Industry Co., Ltd., Tokyo, Japan) dissolved in 50 mM phosphate buffer (pH 7.0) and incubated for up to 8 h.

### 4.2. Primers

Primers used in this study are listed in Appendix A.

### 4.3. Cell-Free Protein Synthesis, Immunoblotting, and AlphaScreen System

The full-length open reading frames (ORFs) of *PBL27*, *CRK2*, *CRK3*, *CPK3*, and *CPK13* were cloned into the Gateway (GW) destination vector pEU-6His-bls-GW (bls; biotin ligation site) or pEU-GW-FALG using the Gateway cloning system (Thermo Fisher Scientific, Waltham, MA, USA). Cell-free protein synthesis and AlphaScreen-based protein–protein interaction assays were carried out according to the methods described previously [29]. For evaluation of the quality of the proteins used, total proteins were subjected to 10% SDS-PAGE and immunoblotted with anti-biotin HRP-linked antibody (Cell Signaling Technology, Beverly, MA, USA) or monoclonal anti-FLAG M2-peroxidase antibody produced by mouse clone M2 (Sigma-Aldrich, St. Louis, MO, USA) (Appendix A). The membranes were soaked with Immobilon Western Chemiluminescent HRP Substrate (Merck Millipore Ltd., Darmstadt, Germany), and the signals were detected with an ImageQuant LAS-4000 imaging system (GE Healthcare, Buckinghamshire, UK).

### 4.4. RNA Isolation, cDNA Synthesis and Quantitative Polymerase Chain Reaction (qPCR)

Approximately 100 mg of leaf tissues were homogenized in liquid nitrogen, and total RNA was isolated and purified using TRI reagent (Molecular Research Center, Inc., Cincinnati, OH, USA) following the manufacturer’s protocol. Single-stranded cDNA was synthesized using ReverTra Ace qPCR RT Master Mix with gDNA Remover (Toyobo, Osaka, Japan), and 0.5 µg of the total RNA was incubated, first, at 37 °C for 5 min for the DNase reaction, and then at 37 °C for 15 min for the RT reaction. Real-time PCR was performed using a CFX Connect real-time PCR detection system (Bio-Rad) with THUNDERBIRD SYBR qPCR Mix (Toyobo) and gene-specific primers (Appendix A). The following protocol was used: an initial polymerase activation of 60 s at 95 °C, followed by 45 cycles of 15 s at 95 °C and then 30 s at 60 °C. Then a melting curve analysis preset by the instrument was performed. Relative transcript abundances were determined after normalization of raw signals with the abundance of the housekeeping transcript of the Arabidopsis *ACT8* gene (at1g49240).

### 4.5. Co-immunoprecipitation Assay and Phosphorylation Assay

The full-length ORFs of *PBL27*, *CRK2*, *CRK3*, *CPK3*, *CPK13*, *PBL27^KD^* (Lys112 to Glu), and *CRK2^KD^* (Lys176 to Arg) were inserted into the GW vector pGWB11 (cauliflower mosaic virus 35S promoter [35SP]::GW::FLAG::*nopaline synthase* terminator [NOST]) or pGWB14 (35SP::GW::3xHA::NOST). *Agrobacterium tumefacience* EHA105 carrying each vector was pressure-infiltrated into the leaves of *N*. *benthamiana*, and the plants were incubated for 2 days. For the co-immunoprecipitation, total proteins were extracted with the extraction buffer (50 mM Tris-HCl (pH 7.5), 150 mM NaCl, 10% glycerol, 5 mM dithiothreitol [DTT], 2 mM ethylenediaminetetraacetic acid [EDTA], 1 mM NaF, 1 mM Na_2_MoO_4_–2H_2_O, 0.5% polyvinylpyrolidone, 1% NP-40, and Complete Protease Inhibitor Cocktail tablets (Roche Applied Science, Indianapolis, IN, USA)). Extracted proteins were incubated overnight with anti-DYKDDDDK-tag antibody magnetic beads (Wako Pure Chemical Industrials, Ltd., Osaka, Japan) at 4 °C, and the beads were washed four times with TBS containing 0.5% NP-40. Immunoprecipitates were eluted with a Laemmli SDS sample buffer and used for 8% SDS-PAGE. For the phosphorylation assay, total proteins were extracted using a Laemmli SDS sample buffer and used for 8% SDS-PAGE, which contained 2.5 µM phos-tag (Wako Pure Chemical Industrials, Ltd.). Anti-FLAG antibody (Sigma-Aldrich) and anti-HA antibody (Roche Applied Science) were used as the primary antibodies. Horseradish peroxidase-linked anti-mouse antibody (Cell Signaling Technology) or anti-rat antibody (Cell Signaling Technology) was used as the secondary antibody. The membranes were soaked with Immobilon Western Chemiluminescent HRP Substrate (Merck Millipore Ltd.) and the signals were detected with an ImageQuant LAS-4000 imaging system (GE Healthcare).

### 4.6. Protoplast Preparation and Transfection

The full-length ORF of *ERF13* was cloned into the p35SΩ-GW-NOST vector [35SP::Ω sequence (translation enhancer)::GW region::NOST) [16]]. The promoter region (1000 bp upstream region) of *PDF1.2* (PDF1.2P) was cloned and inserted in front of the *firefly luciferase* (*Fluc*) reporter gene::NOST cassette in the pMA cloning vector (Thermo Fisher Scientific). Protoplast isolation from Arabidopsis leaves was performed as previously described [30]. Finally, 2.0 × 10^5^ protoplasts mL^−1^ were prepared. Polyethylene glycol-mediated DNA transfection was performed as previously described [31]. The protoplast suspension (100 μL) was supplemented with a mixture of vectors carrying reference (35SP::*Renilla luciferase* [*Rluc*]::NOST), PDF1.2P::*Fluc*::NOST, and 35SP::*ERF13*::NOST at a ratio of 1:4:5 to protoplast suspension with 100 μL PEG solution (40% (*w*/*v*) polyethylene glycerol, 0.4 M mannitol, and 0.1 M Ca(NO_3_)_2_-4H_2_O). The transfection was carried out at room temperature for 15 min and stopped by adding 400 μL of W5 solution (2mM MES (pH 5.7), 5mM KCl, 125mM CaCl_2_, 154mM NaCl). The protoplasts were collected by centrifugation at 100 g for 2 min and resuspended with 500 μL of WI solution (5 mM MES (pH 5.7), 0.4 M mannitol, and 20 mM KCl) and incubated in a 2 mL tube at room temperature overnight.

### 4.7. LUC Assay

The LUC assay was performed as previously described [32]. Fluc activity produced due to the transformed reporter construct was expressed as the value normalized by the Rluc activity produced due to the co-transfected reference vector. Replicate analyses were conducted with 3 independent samples.

### 4.8. Statistics and Reproducibility

We performed Student’s *t*-test for pairwise analysis and one-way ANOVA with Holm’s sequential Bonferroni post hoc test using the program (http://astatsa.com/OneWay_Anova_with_TukeyHSD/ (accessed in 1 April 2023)) for comparing multiple samples. The sample sizes and number of replicates for all of the sets of assays and analyses are indicated in the legends of the corresponding figures.

## Figures and Tables

**Figure 1 plants-12-01747-f001:**
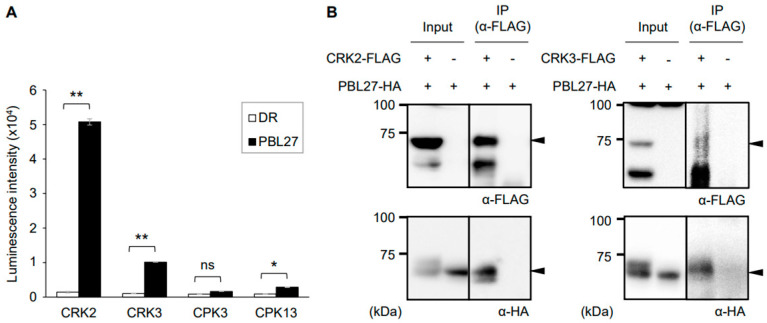
Interactions of PBL27 and cytoplasimic kinases. (**A**) Luminescence intensities are based on the AlphaScreen assay to assess the interactions between biotinylated (Bio)-proteins for PBL27 and FLAG-conjugated proteins for CRK2, CRK3, CPK3, CPK13, with *Escherichia coli* dihydrofolate reductase (DR) serving as control. Data presented in the bottom panel represent the mean and standard error (*n* = 3). Recombinant proteins synthesized using the cell-free system are presented in Appendix A. Data marked with an asterisk(s) are significantly different based on a student’s *t*-test (* 0.01 ≤ *p* < 0.05; ** *p* < 0.01). ns, not significant. (**B**) A pair of FLAG-tagged CRK2 or CRK3 (CRK2−FLAG or CRK3−FLAG) and HA-tagged PBL27 (PBL27−HA) were expressed in *Nicotiana benthamiana* leaf cells. Total proteins extracted from the leaves were immunoprecipitated using anti-FLAG-tag magnetic beads, subjected to SDS-PAGE, and probed with the respective antibodies (α−HA or α−FLAG) as a primary antibody. Arrowheads indicate the predicted target signals. IP, immunoprecipitation.

**Figure 2 plants-12-01747-f002:**
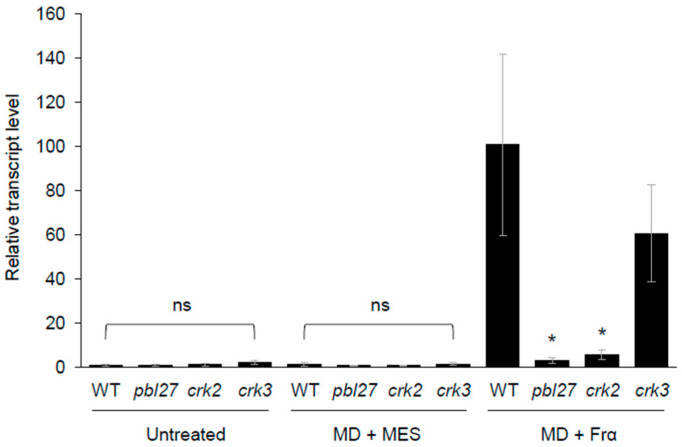
In planta transcriptional response of *PDF1.2* to Frα. Leaves of wild-type (WT) Arabidopsis and *pbl27*, *crk2*, and *crk3* mutant plants were subjected to mechanical damage (MD) with application of MES buffer or Frα, and transcript levels of *PDF1.2* in leaves after 24 h were analyzed. Data represent the mean and standard error (*n* = 5–6). Data marked with an asterisk are significantly different from those of WT leaves in each dataset (untreated, MD + MES, and MD + Frα) based on an ANOVA with Holm’s sequential Bonferroni post-hoc test (* 0.01 ≤ *p* < 0.05). ns, not significant.

**Figure 3 plants-12-01747-f003:**
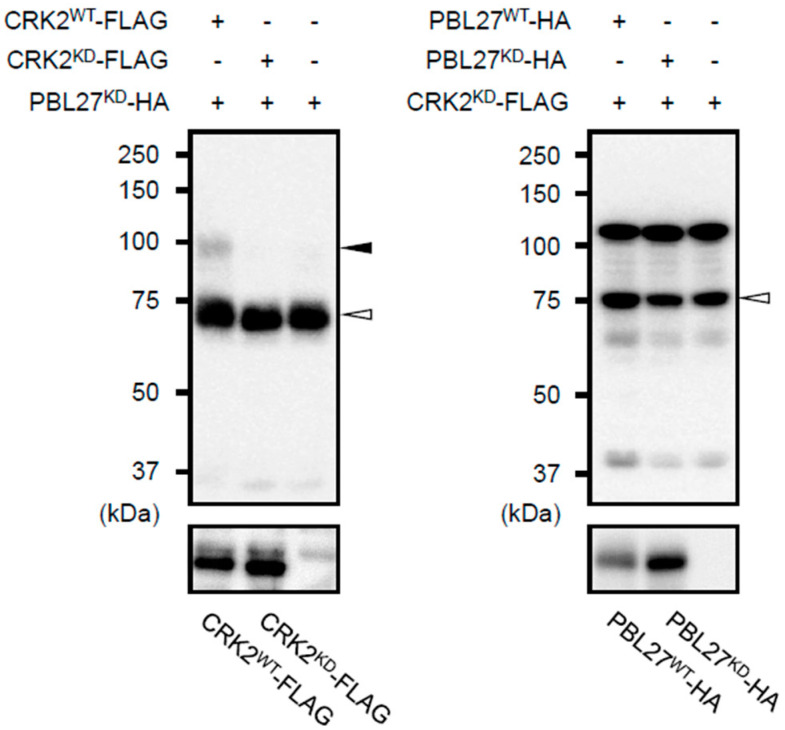
CRKs phosphorylate PBL27 in vitro. Kinase active/dead FLAG-tagged CRK2 (CRK2^WT^−FLAG or CRK2^KD^−FLAG), kinase dead HA-tagged PBL27 (PBL27^KD^−HA) or kinase active/dead HA-tagged PBL27 (PBL27^WT^−HA or PBL27^KD^−HA), and kinase dead FLAG-tagged CRK2 (CRK2^KD^−FLAG) were coexpressed in *Nicotiana benthamiana* leaf cells. Total proteins extracted from the leaves were subjected to SDS-PAGE and probed with the respective antibodies (α−HA or α−FLAG) as a primary antibody (lower panel). Phosphorylation was detected by a mobility shift on a phos-tag SDS-PAGE gel (upper panel). White arrowheads indicate the predicted target signals, Black arrowheads indicate the mobility-shifted target signal.

**Figure 4 plants-12-01747-f004:**
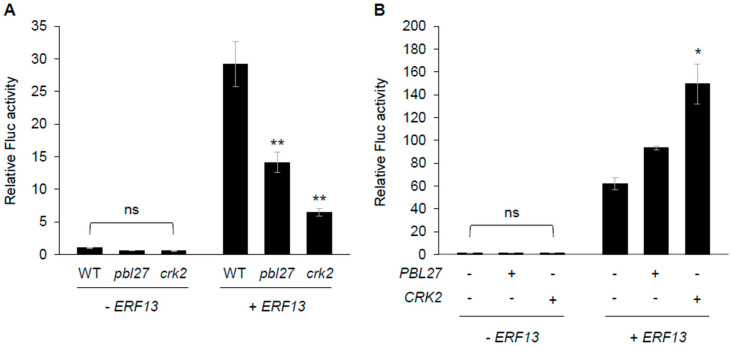
Modulation of ERF13-mediated *PDF1.2* promoter (PDF1.2P) activities by PBL27 and CRK2. Transient activation of a *firefly luciferase* reporter gene under PDF1.2P according to coexpression with (+) or without (−) *ERF13* in protoplasts prepared from leaves of Arabidopsis wild-type (WT) and PBL27- and CRK2-deficient mutant plants (*pbl27* and *crk2*, respectively) (**A**) and coexpression with (+) or without (−) *ERF13* and either *PBL27* or CRK2 in WT leaf protoplasts. Data represent the mean and standard error (*n* = 4 for (**A**) and *n* = 3 for (**B**)). Data marked with an asterisk(s) are significantly different from those of WT (**A**) or those obtained without *PBL27* and *CRK2* (**B**) among each data set based on an ANOVA with Holm’s sequential Bonferroni post-hoc test (* 0.01 ≤ *p* < 0.05; ** *p* < 0.01). ns, not significant.

**Figure 5 plants-12-01747-f005:**
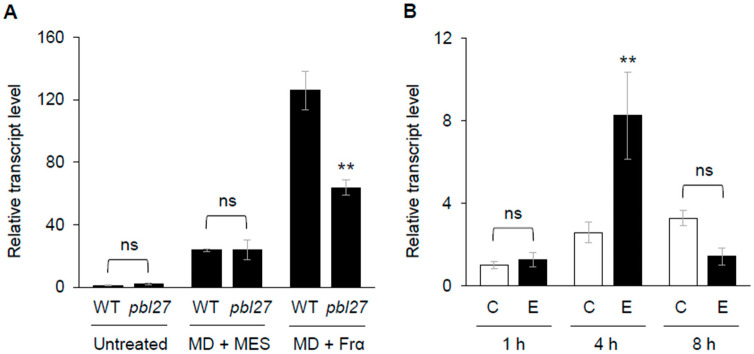
The PBL27-mediated ethylene signaling for upregulation of *ERF13* transcript. (**A**) Leaves of wild-type (WT) Arabidopsis and *pbl27* mutant plants were subjected to mechanical damage (MD) with application of MES buffer or Frα. Plants were incubated for 1 h. (**B**) Leaves of WT plants were treated with ethephon [E] or phosphate buffer (control [C]). Plants were incubated for up to 8 h. Transcript levels of *ERF13* in leaves were analyzed. Data represent the mean and standard error (*n* = 4–5). Data marked with asterisks are significantly different from those of WT leaves in each dataset (untreated, MD + MES, and MD + Frα) (**A**) or those with control treatment (**B**) based on an ANOVA with Holm’s sequential Bonferroni post-hoc test (** *p* < 0.01). ns, not significant.

**Figure 6 plants-12-01747-f006:**
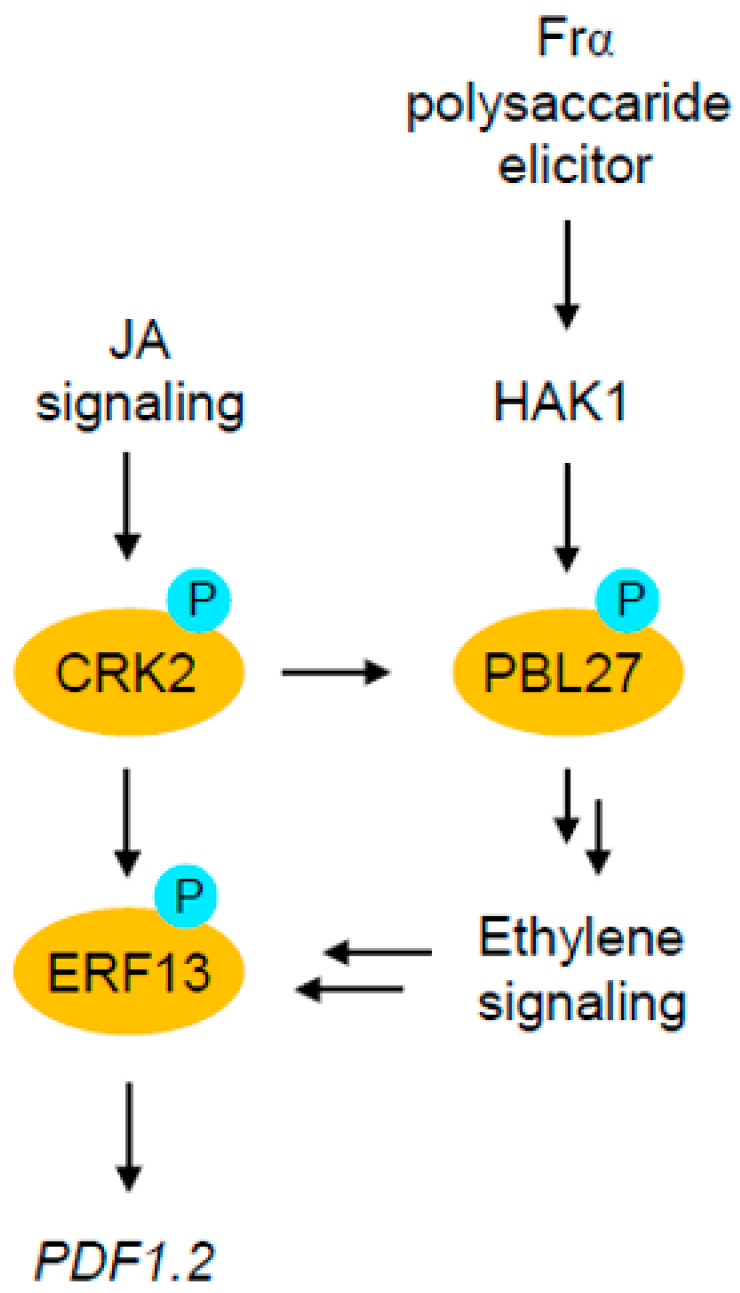
A possible model of the polysaccharide elicitor (Frα)-induced activation of the cellular signaling network for *PDF1.2* expression in Arabidopsis in response to *S*. *litura* attack. Note that the Frα-responsive phosphorylation of PBL27 with CRK2 remains to be explored.

## Data Availability

The data that support the findings of this study are available from the corresponding author upon reasonable request.

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
