# Peer review of "Cytoplasmic Kinase Network Mediates Defense Response to Spodoptera litura in Arabidopsis"

_plants, 2023, doi:10.3390/plants12091747_

Round 1

Reviewer 1 Report

The manuscript presented by Desaki et al. show that CRK2 regulates ERF13 phosphorylation and PBL27-dependent de novo synthesis of ERF13, thus determining active defense traits against S. litura larvae via transcriptional regulation of PDF1.2. The new discoveries in this story include: 1) PBL27 interacts with CRK2 abd CRK3, and CRK2 phosohorylates PBL27; 2) the MD+Frα induced expression of PDF1.2 was disrupted in pbl27 and crk2 mutant plants; 3). The ERF13-mediated expression of PDF1.2 was weaken in pbl27 and crk2 mutant plants; 4). PBL27-mediated ethylene signaling induced ERF13 expression. These findings are interesting but are hampered by a number of inconsistencies and missing controls in the experiments. However, there are quite a few issues that I think the authors need to address. 

1. In figure 1A, the best control is homologous gene of PBL27, but not DR in E. coli. In figure 1B, the signal of CRK3-FLAG from IP(anti-FLAG) is blurry, it is better to provide a better picture. 

2. In lines104-105, the result of PDF1.2 transcript level assay is not enough for the conclusion "the PBL27/CRK2 system is ultimately responsible for the S. litura elicitor response", I think it can be revised to "the S.litura-induced PDF1.2 expression is dependent on PBL27/CRK2". 

3. How do you think about the relationship between PBL27-CRK2 and PBL27-CRK3? PBL27 interacts with CRK2 and CRK3 (figure1), but CRK2 and CRK3 show different function in S. litura-induced PDF1.2 expression. 

4. In figure 3, the kinase CRK2-FLAG and PBL27-HA should be added as control respectively, and in the right panel, how do you rule out the possibility that the upper bands (above 100 kD) is masking the phosphorylated band of CRK2. 

5. How do you think about the difference of relative Fluc activity in WT+ERF13, in figure 4A the data is nearly 30, however, in figure 4B, the data is about 60. 

6. In figure 5, there is not enough evidence for "the PBL27-mdeiated ethylene signaling is upstream of ERF13 transcript", the ERF13 expression level should be tested in pbl27 mutant plants under ethephon treatment. 

A major weakness of the manuscript is the writing. Besides grammatical errors, there are quite many overstatements, confusing sections, and many parts could be shorten and integrated (some problematic areas are marked in yellow in the modified manuscript). Also, the discussion could be further developed.

Author Response

All the comments from you were very insightful and useful and have improved the quality of our manuscript after we fully responded to them.

  1. In figure 1A, the best control is homologous gene of PBL27, but not DR in E. coli. In figure 1B, the signal of CRK3-FLAG from IP(anti-FLAG) is blurry, it is better to provide a better picture. 

Response: We are not sure whether the PBL27 homologue is not really able to interact CRKs, as many of cytoplasmic kinases have redundant functions. I feel that in vivo data in Figure 1B confirms the findings on the AlphaScreen assay.

Response: The photo used for the analysis of CRK3-FLAG from IP(anti-FLAG) is the best-qualitied one. The intensity of the regarding band is very week, and this would be consistent with the data in Fig. 1A. 

  1. In lines104-105, the result of PDF1.2 transcript level assay is not enough for the conclusion "the PBL27/CRK2 system is ultimately responsible for the S. litura elicitor response", I think it can be revised to "the S.litura-induced PDF1.2 expression is dependent on PBL27/CRK2". 

Response: We rewrote the sentence according to your comments. Thank you.

  1. How do you think about the relationship between PBL27-CRK2 and PBL27-CRK3? PBL27 interacts with CRK2 and CRK3 (figure1), but CRK2 and CRK3 show different function in S. litura-induced PDF1.2 expression. 

Response: As described above, CRK3 is not likely to interact strongly with PBL27. Therefore, we think that CRK3 is not aggressively involved in the S. litura-indued defense response. Please see the Discussion section.

  1. In figure 3, the kinase CRK2-FLAG and PBL27-HA should be added as control respectively, and in the right panel, how do you rule out the possibility that the upper bands (above 100 kD) is masking the phosphorylated band of CRK2. 

Response: We are sorry, but we did not explain about the bottom panel serving as control in Figure 3. We added the following in the figure legend. We also considered masking by unspecific bands and analyzed CRK2KD-FLAG band-sift using 4-fold concentration (10 mM) of phos-tag gel. However, we could not detect any band sift.

  1. How do you think about the difference of relative Fluc activity in WT+ERF13, in figure 4A the data is nearly 30, however, in figure 4B, the data is about 60. 

Response: Variable efficiencies on gene delivery in each set of reporter assays make difference on the relative thresholds of data. 

  1. In figure 5, there is not enough evidence for "the PBL27-mdeiated ethylene signaling is upstream of ERF13 transcript", the ERF13 expression level should be tested in pbl27 mutant plants under ethephon treatment. 

Response: Because ethylene is considered to act downstream of PBL27, treatment of WT and pbl27 mutants with ethephon (ethylene) may not result in a clear difference in ERF13 expression between them. In any case, we toned-down the description in the regarding discussion in the Discussion section.

A major weakness of the manuscript is the writing. Besides grammatical errors, there are quite many overstatements, confusing sections, and many parts could be shorten and integrated (some problematic areas are marked in yellow in the modified manuscript). Also, the discussion could be further developed.

Response: Thank you again for these valuable comments. We carefully checked and revised our manuscript.

Reviewer 2 Report

The study of Yoshitake Desaki et al. “Cytoplasmic kinase network mediates defense response to 3 Spodoptera litura in arabidopsis” provided clear evidence that Arabidopsis response to Spodoptera litura larval elicitors involves CRK2-regulated phosphorylation of ERF13 and PBL27-dependent de novo synthesis via transcriptional regulation PDF1.2.

The paper is well presented and the English is clear and easily readable. It has a clearly structured and informative introduction, clearly stated goals, and clearly defined and appropriate methodology. The discussion is appropriate to the topic.

Minor comments:

- Please proof read carefully to find out typos, a space, or an altered font in the main text.

- Describe what is shown on the bottom panel of Figure 3.

- My guess is that Arabidopsis is more used to capital letters.

Author Response

Reviewer2

- Please proof read carefully to find out typos, a space, or an altered font in the main text.

Response: We checked them throughout our manuscript. Thank you.

- Describe what is shown on the bottom panel of Figure 3.

Response: We added the description.

- My guess is that Arabidopsis is more used to capital letters.

Response: We revised it.

Reviewer 3 Report

This work revealed that CRKs functioned with PBL27 in the intracellular signaling network for anti-herbivore responses. And further results indicated that the CRK2/PBL27 system is predominantly responsible for the Frα-responsive transcription of PDF1.2 in S. litura-damaged plants. At last, results in this work show that CRK2 regulates not only ERF13 phosphorylation but also PBL27-dependent de novo synthesis of ERF13, thus determining active defense traits against S. litura larvae via transcriptional regulation of PDF1.2. The results provided in this research provided new insight into the complex function of CRK2 that coordinates with PBL27 for resultant transcriptional activation of PDF1.2 in response to a polysaccharide elicitor (Frα) during S. litura attack. However, how CRK2 interacts with PBL27 remains unclear in the working model. More experiments, such as one/two yeast hybrid and EMSA should be carried out in the future to explore the key genes in the interacting process between CRK2 and PBL27.

Author Response

More experiments, such as one/two yeast hybrid and EMSA should be carried out in the future to explore the key genes in the interacting process between CRK2 and PBL27.

Response: We added the following sentences in the Discussion section. The molecular mechanism by which phosphorylation of ERF13 by the complex of CRK2 and PBL27 activates ERF13transcription remains unclear. To reveal how protein phosphorylation has significant effects on the accurate transcript machinery, Further analysis of the interaction between the complex and the ERF13 promoter using electrophoretic mobility shift assays, etc. will be required.